# Radiothermal Emission of Nanoparticles with a Complex Shape as a Tool for the Quality Control of Pharmaceuticals Containing Biologically Active Nanoparticles

**DOI:** 10.3390/pharmaceutics15030966

**Published:** 2023-03-16

**Authors:** Anton V. Syroeshkin, Gleb V. Petrov, Viktor V. Taranov, Tatiana V. Pleteneva, Alena M. Koldina, Ivan A. Gaydashev, Ekaterina S. Kolyabina, Daria A. Galkina, Ekaterina V. Sorokina, Elena V. Uspenskaya, Ilaha V. Kazimova, Mariya A. Morozova, Varvara V. Lebedeva, Stanislav A. Cherepushkin, Irina V. Tarabrina, Sergey A. Syroeshkin, Alexander V. Tertyshnikov, Tatiana V. Grebennikova

**Affiliations:** 1Department of Pharmaceutical and Toxicological Chemistry, Medical Institute, RUDN University, 6 Miklukho-Maklaya Street, 117198 Moscow, Russia; 2Department of Environmental Instrumentation, National Technical University of Ukraine “Igor Sikorsky Kyiv Polytechnic Institute”, 37, Prosp. Peremohy, 03056 Kyiv, Ukraine; 3I.I. Mechnikov Research Institute for Vaccines and Sera, 105064 Moscow, Russia; 4Federal Government Budgetary Institution “National Research Center for Epidemiology and Microbiology Named after Honorary Academician N.F. Gamaleya” of the Ministry of Health of the Russian Federation, 18 Gamaleya St., 123098 Moscow, Russia; 5E. K. Fedorov Institute of Applied Geophysics, 129128 Moscow, Russia

**Keywords:** nanoparticles, supramolecular structures, thermal radio emission, drug quality control

## Abstract

It has recently been shown that the titer of the SARS-CoV-2 virus decreases in a cell culture when the cell suspension is irradiated with electromagnetic waves at a frequency of 95 GHz. We assumed that a frequency range in the gigahertz and sub-terahertz ranges was one of the key aspects in the “tuning” of flickering dipoles in the dispersion interaction process of the surfaces of supramolecular structures. To verify this assumption, the intrinsic thermal radio emission in the gigahertz range of the following nanoparticles was studied: virus-like particles (VLP) of SARS-CoV-2 and rotavirus A, monoclonal antibodies to various RBD epitopes of SARS-CoV-2, interferon-α, antibodies to interferon-γ, humic–fulvic acids, and silver proteinate. At 37 °C or when activated by light with λ = 412 nm, these particles all demonstrated an increased (by two orders of magnitude compared to the background) level of electromagnetic radiation in the microwave range. The thermal radio emission flux density specifically depended on the type of nanoparticles, their concentration, and the method of their activation. The thermal radio emission flux density was capable of reaching 20 μW/(m^2^ sr). The thermal radio emission significantly exceeded the background only for nanoparticles with a complex surface shape (nonconvex polyhedra), while the thermal radio emission from spherical nanoparticles (latex spheres, serum albumin, and micelles) did not differ from the background. The spectral range of the emission apparently exceeded the frequencies of the Ka band (above 30 GHz). It was assumed that the complex shape of the nanoparticles contributed to the formation of temporary dipoles which, at a distance of up to 100 nm and due to the formation of an ultrahigh strength field, led to the formation of plasma-like surface regions that acted as emitters in the millimeter range. Such a mechanism makes it possible to explain many phenomena of the biological activity of nanoparticles, including the antibacterial properties of surfaces.

## 1. Introduction

Most substances with medicinal properties are receptor ligands or enzyme inhibitors. Apart from the multipoint Coulomb binding, which is well known for low-molecular-weight substances, supramolecular drugs (including biopolymers [1,2,3]) interact with a supramolecular target due to “surface-to-surface” dispersion (Deryagin) interactions [4]. A specific feature of these interactions is the Coulomb tuning of the flickering dipoles due to the mutual electromagnetic radiation of the surfaces [5]. The contribution of dispersion interactions to the free energy of a surface’s adhesion reaction is an order of magnitude larger than the interaction of single charges [6]. Therefore, the dynamic redistribution of surface dipoles when a supramolecular drug binds to a supramolecular receptor is a critical condition in the mechanism of the biological activity of nanoparticles. The electromagnetic radiation of flickering dipoles on the surface of nanoparticles (including the polarization of the entire nanoparticle) is expected to be non-anisotropic, taking into account Brownian motion. Under this assumption, radio engineering devices that generate electromagnetic radiation in the millimeter range can have a significant effect on the ligand–receptor interaction if the ligand is a supramolecular structure. Indeed, electromagnetic radiation at a frequency of 95 GHz completely blocked the infectious activity of the SARS-CoV-2 virus [7]. It was assumed that it would be possible to register an intrinsic thermal radio emission from polarized, biologically active nanoparticles at their activation. Indeed, the radiation of nanoparticles of complex shapes (for example, antibodies, interferons, and virus-like particles (VLP)) in the microwave range is at least an order of magnitude stronger than the background after they are heated or irradiated with light. This article is dedicated to the description of this phenomenon, bearing in mind that the thermal radio emission property of nanoparticles can be used to control the quality of drugs. When selecting nanoparticles, it is important to pay attention to the drugs required to resist the novel coronavirus infection and its concurrent mixed viral infections. For this reason, the list of studied nanoparticles includes monoclonal antibodies to SARS-CoV-2 [8], VLPs to SARS-CoV-2 and the rotavirus [9,10,11], humic–fulvic acids [12], and preparations of interferons and antibodies to these viruses. [13,14].

## 2. Materials and Methods

### 2.1. Nanoparticles

The preparation of monoclonal antibodies (mAb) to the SARS-CoV-2 RBD (receptor-binding domain) was achieved in the manner described below. Cell strains (specific cell strains) producing monoclonal antibodies to the receptor-binding domain of the S-protein of the SARS-CoV-2 virus were obtained using hybridoma technology. Balb/c mice weighing 18–20 g were immunized three times with the recombinant receptor-binding domain of the S-protein of the SARS-CoV-2 virus. The antigen was obtained in a Chinese hamster ovary (CHO) cell culture. The interval between immunizations was 2 weeks. Animals were injected with the antigen intraperitoneally (100 μg/mouse) with Freund’s complete adjuvant (first immunization) or Freund’s incomplete adjuvant (second and third immunizations). A booster dose (100.0 µg/mouse) was administered intraperitoneally without an adjuvant 3 days before spleen isolation.

Three days after the injection of the booster dose of the antigen, a suspension of mouse spleen cells was obtained by the standard method in a cooled growth medium without serum [3].

A continuous cell line of mouse myeloma Sp2/0-Ag-14, which was defective in the HGPRT gene and did not produce its own immunoglobulins, was used for fusion. The hybridization was carried out according to the classic Milstein and Köhler technique. The suspension of spleen cells and myeloma cells were mixed in the ratio of 5:1, respectively. After fusion, the cells were resuspended in the selective medium with the addition of the Hypoxanthine-Aminopterin-Thymidine (HAT) media supplement (Hybri-Max^®^, Sigma, St. Louis, MO, USA) and transferred into 96-well plates with a feeder layer of mouse peritoneal macrophages (5 × 10^6^ cells/mL, 50 μL/well) at a concentration of 1.5 × 10^5^ cells per well in a volume of 0.1 mL. The plates with cells were incubated at 37 °C in a 5% CO_2_ atmosphere.

The medium in the wells containing clones of the hybridoma cells was replaced every 2–3 days using the medium with HAT within 14 days after fusion. To select a hybridoma producing a mAb of a given specificity, culture liquids were taken from the wells with growing colonies. When clones of the hybrid cells occupied at least 50% of the well surface, they were tested for antibodies to the receptor-binding domain of the S-protein of the SARS-CoV-2 virus in an enzyme-linked immunosorbent assay. Clones that demonstrated specific activity were re-cloned using the limiting dilution technique in 96-well plates with a feeder layer of mouse peritoneal macrophages (5 × 10^6^ cells/mL, 50 μL/well). Supernatants from the wells containing re-cloned hybridomas were again tested for the presence of antibodies and their specificity. Positive, re-cloned hybridomas were grown in culture flasks of increasing size (first 25 cm^2^, then 75 cm^2^) (Greiner, Frickenhausen, Germany).

The isolation of the mAb from the culture liquids (CLs) was performed using an AKTA START chromatography system (GE Healthcare, Uppsala, Sweden) and a 1.0 mL HiTrap Protein A HP column (GE Healthcare, Sweden), according to the manufacturer’s protocol. Desalting and replacing the buffer in the sample purified at the first stage were achieved using an AKTA START chromatography system (GE Healthcare, Sweden) and a 5 mL HiTrap desalting column (GE Healthcare, Sweden), according to the manufacturers’ protocols.

There were four clones of the hybrid cells selected (Table 1) which demonstrated the best ELISA results when interacting with the recombinant antigen. One of the clones produced monoclonal antibodies that neutralized the SARS-CoV-2 virus in tests in vitro.

The constant of association with the receptor-binding domain of the S-protein for the virus-neutralizing antibodies produced by clone 2E1 was determined using an Octet Red 96 molecular interaction detection system (ForteBio, Fremont, CA, USA) with Ni-NTA (NTA) Dip and Read Biosensors (ForteBio, USA), according to the manufacturer’s recommendations. The average association constant (KD) = 8.2 × 10^−9^ (±1.6 × 10^−9^) M.

The preparation of virus-like particles (VLP) mimicking rotavirus A was achieved as described in [15].

Virus-like particles (VLP) mimicking SARS-CoV-2 were obtained and purified as described below.

Virus-like particles were obtained by self-assembly from recombinant proteins synthesized in the baculovirus expression system. Recombinant baculoviruses carrying the genes of four proteins (S, E, M and N) of SARS-CoV-2 were obtained, and the S protein genes were obtained from the most common, relevant clades in the Russian Federation and Europe: Wuhanese, South African, British, and Indian. The selected amino acid sequences were optimized for production in insect cells and cloned into the pFastBacHTa transfer vector. The continuous *Spodoptera frugiperda* (Sf-21) cell line was transfected with purified bacmid DNA preparations (bac-RVA) using the cationic liposomal agent Cellfectin (Invitrogen, USA). Two clones were used for each construct (a culture concentration of cells of 5 × 10^5^/mL and 10 µL of bacmid). After transfection, two more passages were carried out on Sf-21 cells, followed by co-infection with a continuous culture of *Trichoplusia ni* insect cells, which was cultivated for 7 days after infection.

The culture liquid containing the virus-like particles was subjected to low-speed centrifugation, removing cells and cell debris at 1000 rpm for 5 min and at 6000 rpm for 20 min, respectively (4 °C, Sorval^®^ SS34 rotor). The resulting clarified suspensions were layered on 6 mL of 25% or 35% (*w*/*v*) sucrose prepared in TNE buffer (50 mM Tris + 100 mM NaCl, 0.5 mM EDTA, pH = 7.4)). They were centrifuged for 2 h at 28,000 rpm (Optima XE-100 centrifuge, SW 32Ti rotor, Beckman Coulter, 4 °C). The obtained pellets were re-suspended in TNE buffer and stored at 4 °C.

Recombinant interferon-α-2b (INFα) [16] was obtained as a component of the registered drug (№ P 001503 RU) in the form of a solution.

Antibodies to interferon-γ (AINF-γ) (as a component of registered drug № P (000023)-(RG-RU)-140422) were obtained in the fluidized bed aerosol chamber in the form of a powdered substance (in a carrier) saturated with hydroalcoholic AINF-γ. The characteristics of the substance were previously described [17]. Placebo preparations of antibodies to INF-γ were also specially made in the aerosol chamber. The AINF-γ solutions were replaced by phosphate-buffered saline before being placed in the fluidized bed chamber [17,18]. The carrier—an intact powder of lactose monohydrate used for the manufacture of a pharmaceutical substance—was also investigated.

The preparation of humic–fulvic acid was received from the Sistema Biotechnologii company as previously described [19].

Bovine serum albumin (in a lyophilized powder form) was obtained from Sigma-Aldrich (St. Louis, MO, USA).

### 2.2. Indirect Enzyme-Linked Immunosorbent Assay (ELISA)

To assess the efficiency of mAb binding to RBD, the recombinant RBD protein was adsorbed on microplate wells overnight at 4 °C in 0.1 M carbonate buffer at a pH of 9.5 and a concentration of 2 μg/mL in 100 μL per well. After washing the plate four times, 100 μL of purified mAbs from each clone was added to the wells at concentrations from 0.02 to 50 μg/mL. The plate was incubated for 1 h at 37 °C. After washing the plate, 100 µL of mAb-based peroxidase conjugated to mouse IgG was added to the wells. The plate was incubated for 1 h at 37 °C. After washing the plate, 100 µL of a substrate solution with tetramethylbenzidine was added. The plate was incubated for 15 min at room temperature in the dark. Then, 100 µL of 1 M H_2_SO_4_ was added to stop the reaction. The optical density at 450 nm (A_450_) was measured using a Multiscan EX vertical beam spectrophotometer (Thermo, Waltham, MA USA).

### 2.3. Detection of the Receptor-Binding Domain (RBD) of S-Protein in Sandwich ELISA Format

For the detection of RBD in the sandwich ELISA format, purified 2E1B5 mAbs were adsorbed on microplate wells overnight at 40 °C in 0.1 M carbonate buffer at a pH 9.5 and a concentration of 10 μg/mL, 100 μL per well. After washing the plate four times, 100 μL of RBD was added to the wells at concentrations ranging from 0.01 to 5000 μg/mL. The plate was incubated for 1 h at 37 °C. After washing the plate, 100 µL of 2E1B5 mAb peroxidase conjugate was added to the wells. The plate was incubated for 1 h at 37 °C. After washing the plate, 100 µL of substrate solution with tetramethylbenzidine was added. The plate was then incubated for 15 min at room temperature in the dark, and 100 µL of 1 M H_2_SO_4_ was added to stop the reaction. A_450_ was measured using a Multiskan EX vertical beam spectrophotometer (Thermo, USA).

### 2.4. Method for Determining the Size of Nanoparticles

The size distribution of the nanoparticles and the control of their monodispersity were determined using a ZetasizerNano ZS dynamic light scattering spectrometer (DLS) (MALVERN Instruments, Malvern, UK). The applications of this device include the measurement of diluted samples and nanoparticle size distribution from 0.1 nm to 10 µm using patented NIBS (Non-Invasive Back Scattering) technology. The measurement principle was based on photon correlation spectroscopy, which is based on the analysis of the Brownian motion of particles in the dispersed phase in a dispersion medium, which leads to fluctuations in the local concentration of particles, the local heterogeneities of the refractive index, and the intensity of scattered light. The characteristic relaxation time of intensity fluctuations is inversely proportional to the diffusion coefficient. Particle size was calculated using the Stokes–Einstein equation:(1)$$D=\frac{k_{B}T}{6\pi \eta r}$$
where *D* is the diffusion coefficient, *k_B_* is the Boltzmann constant, *T* is the absolute temperature, *η* is the viscosity of the liquid, and *r* is the radius of the particle.

Each solution of nanoparticles was controlled for monodispersity using the DLS method and the absence of impurities in the range of more than 190 nm. The buffer solutions and high-resistivity water (Millipore, Burlington, MA, USA) were also monitored for impurities.

### 2.5. The Density of the Thermal Radio Emission Flux

The density of the thermal radio emission flux in the microwave range of wavelengths was determined using a TES92 apparatus (TES Electrical Electronic Corp., Taipei, Taiwan) with the device tuned for anisotropic measurement along the *Z* axis. The measurement results were recorded as the maximum average value of the flux density at a given point in time with stepwise averaging every 300 ms.

The powders or solutions were heated using a solid-state thermostat with Peltier elements. Control over the temperature of the sample maintained using a remote laser infrared thermometer.

The samples were activated by light with new-generation LEDs which provide a power density of up to 50 mW/cm^2^ in the region of 412 nm at a spectral line width of up to 2–4 nm (including model AA3528LVBS/D, type C503B-BCN-CV0Z0461, CreeLED, Durham, NC, USA).

All measurements were carried out on the unit in strictly the same geometry, located in a room where the second TES92 device controlled the distribution of the microwave background radiation with a height, width, and length pitch of 50 cm. The background radiation in the experimental room did not exceed 1 μW/m^2^ at all monitoring points.

Aqueous solutions were applied in drops of 100 µL to the center of the bottom of sterile 10 cm Petri dishes. The drops were applied axially along the *Z* axis, 10 mm from the measuring head of the device. The powders were poured in doses of 30 g into 3 cm sterile Petri dishes with uniform shrinkage.

All measurements were taken at least seven times. The standard deviation is shown in the graphs of the Section 3. TES92, as an electric field meter, was calibrated with a relative error of 1 dB and had a low temperature error (0.2 dB in the range from 0 °C to 50 °C).

### 2.6. Spectral Characteristics

The spectral characteristics of the thermal radio emission of the powdered substances were obtained using a special conical antenna with a preamplifier on the spectrum analyzer from the Agilent company. The recorded frequency range was 68 GHz–111 GHz. The scanning pitch over the spectrum was 300 MHz. The diameter of the horn antenna was matched to a 10 cm Petri dish. Heating and temperature were controlled in the same way as in the device described earlier in Section 2.5.

### 2.7. Determination of Protein Concentration

The protein concentration in the solutions was determined using a commercial Micro BCA Protein Assay Kit (Thermo, USA).

### 2.8. Electron Microscopy

To visualize the VLPs, 3 μL of the drug at a concentration of 12–30 mg/mL was applied to a copper grid coated with a carbon substrate (Ted Pella, Redding, CA, USA). It was then treated in the glow discharge atmosphere and incubated for 30 s at room temperature. A drop of 2% uranium acetate solution was then applied to the grid and left for 30 s. The excess solution was then removed from the grid with filter paper. The dyed grids were stored in plastic containers before use. The samples were investigated using a JEOL 2100 transmission electron microscope (JEOL Ltd., Tokyo, Japan) equipped with a lanthanum hexaborite cathode at an accelerating voltage of 200 kV. The images were obtained with a magnification of ×25,000 using a Gatan X100 CCD camera with a matrix size of 2000 × 2000 pixels (Gatan, Inc., Pleasanton, CA, USA). The results for the two types of VLP preparation are shown in Figure 1.

## 3. Results

### 3.1. The Thermal Radio Emission of Nanoparticles Is Stimulated by Heating and Lighting

It is known that the Stefan–Boltzmann equation cannot be used to analyze the emission of nanoparticles as this classical formula provides significantly overestimated results for particles smaller than the wavelength. For example, for particles with sizes r < 100 nm, it was established [20] (based on a combination of Kirchhoff’s law and Mie theory), that the intensity of their thermal radiation, I, is normalized to black body radiation, I_b_
I/I_b_ = (80kT/(π^3^ c ħ)) r J(p)(2)
where J(p)~1/p, the nondimensional parameter p = (r/c) (2π σ k T/ħ)^0.5^, k represents the Boltzmann constant, σ represents the substance conductivity, and c represents the speed of light.

For nanoparticles of copper, W~r^3^T^5^ was determined to be the intensity of their thermal radiation. In W/W_SB_ ≪ 1, W_SB_ is the maximum power of thermal radiation in a vacuum for a macroscopic body limited by the Stefan-Boltzmann law.

It was found that the thermal radio emission of solutions of a number of nanoparticles in solution and amorphous powders actually increases upon heating (by up to three times as the temperature rises from 23 to 37 °C, up to flux density values of 125 μW/m^2^ (~20 mW/(m^2^ sr)). This would be expected based on Formula (1). It was surprising that the power density of the flux from a number of nanoparticles of various natures and sizes (virus-like particles (VLPs) of rotavirus, interferon α (ΙNF-α), antibodies to interferon γ (AINF-γ), and humic–fulvic nanoparticles (“humic acids”)) exceeded the background thermal radio emission by more than an order of magnitude (Figure 2). As a control for possible errors in the TES measurement unit, we measured the emission from latex nanoparticles (LS) and polypeptide bovine serum albumin (BSA) nanoparticles, which was slightly different from the ambient background. Such an excessive thermal radio emission, compared to the background, can result from the oscillating dipole induced on a nanoparticle in the aqueous solution or an amorphous powder with a moisture content of 5–8% (AINF-γ and BSA preparations). The nanoparticle can obtain energy for emission above the background due to its interaction with solvent, leading to the formation of a temporary dipole; for example, the electrolytic dissociation of ionogenic groups (for example, -COOH), the protonation of amino groups, etc. The rate of spontaneous emissive electric dipole decay on the nanoparticle (A_nano_) will then be as shown in [21]:A_nano_ = n_eff_ (E_loc_/E)^2^A_0_(3)
where A_0_ is the rate of electric dipole decay in a vacuum, E_loc_ and E are the microscopic and macroscopic values of the electric field strength on the nanoparticle, respectively, and n_eff_ is the effective refractive index of the nanoparticle substance.

An activated, emitting nanoparticle can be considered an emitting nanoantenna that converts the absorbed power (P) into a microwave radio signal. According to [22]:P = (ω/2) Im{|**p E**_loc_|^2^}(4)
where **p** is the absorption dipole moment and ω is the circular frequency.

The increase in absorption is determined by the magnitude of the local field enhancement, δ = ǀ**E_loc_**ǀ/ǀ**E**ǀ. In the experiment, it achieved values of up to 10^6^ [23]. It is clear that not only does the temperature influence the δ value but also a different influx of energy. In the present work, Figure 2 shows that exposure to light at λ = 412 nm also stimulated thermal radio emission. This was especially effective for the solution of humic–fulvic nanoparticles, since this preparation exhibited good light absorption in this range [19].

### 3.2. The Thermal Radio Emission of Nanoparticles with a Complex Surface Shape (Nonconvex Polyhedra) Significantly Exceeds the Background Values

An increase in the radiation of a nanoantenna can be achieved with a non-spherical shape. This ensures better polarizability, and proteolytic reactions can lead to the appearance of long-lived dipoles at distances from a few nanometers to several tens of nanometers. An analog of such a system is described for a carbon nanotube in which the fullerene has a charge of +3e and the electric field strength is 10^6^ V/cm. Undamped oscillations of the fullerene with a frequency of 0.36 THz are then observed for 50–80 ps [24]. In Table 2, data on the thermal radio emission from spherical nanoparticles and nanoparticles with very complex shapes are separated.

These data indicate that the flux density of the thermal radio emission does not depend on the size of the nanoparticles and significantly exceeds the background only if the shape of the nanoparticle has the complex form of a nonconvex polyhedron: “spikes” on the surface of VLPs (Figure 1), a Y-shaped packing of the oligomeric polypeptide in antibodies to SARS-CoV-2 [25], a complex packing of interferon domains [26], many irregularities on the surface of silver proteinate [27], and complex, diverse forms of humic–fulvic acids [19].

The complex packing of the surface explains the significant enhancement of thermal radio emission in nanoparticles of complex shapes in terms of thermodynamics. The increase in the flux density compared to the background is apparently provided by the energy of the Deryagin disjoining pressure, P_D_, the relationship of which with the Gibbs potential is as follows:P_D_ = −(1/A) (∂G/∂x)(5)
where P_D_ is the disjoining pressure, A is the surface area of the interacting surfaces, G is the Gibbs energy of the interaction of the two surfaces, and x is the distance. With an irregular charge distribution and conformational mobility of nanoparticles, P_D_ increases due to the fact that Equation (4) will be the sum of several partial derivatives of G. Such a dependence on the nanoparticle shape, the arrangement of permanent and temporary dipoles on its surface, and conformational mobility suggest that the thermal radio emission from the nanoparticles will be very specific, depending not only on the material of the nanoparticles but also on their conformation.

### 3.3. Specificity of the Thermal Radio Emission of Nanoparticles

First, the specificity of the thermal radio emission of nanoparticle solutions in preparations of monoclonal antibodies to various epitopes of the SARS-CoV2 RBD domain was demonstrated (Table 3). It should be noted that the virus-neutralizing antibodies (2E1) had the maximum emissivity, which did not depend on heating or lighting. Other mAbs responded differentially to heating or lighting; therefore, the ratio of the flux density of the thermal radio emission during exposure (F_t_—heat; F_l_—light) can be used as a kind of diagnostic feature. As follows from Table 3, the ratio of K= F_t_/F_l_ for mAB 2E1 was 1.0, for 1G2 was 0.46, for 2E7 was 2.7, and for 3D3 was 6.1.

It was assumed that this approach would be universal, and it was applied to the AINF powders. The radiothermal radiation (with thermal and light activation) was compared for the drug and its placebo. Moreover, the placebo was specially made, according to our order, in a fluidized bed aerosol chamber where the AINF preparation was also produced (Table 4). It turned out that the degree of increase in the flux density after heating was 1.6 for AINF-γ, 2.3 for the placebo, and 1.1 for the inert carrier (lactose). The moisture content of these preparations was 8%, and the removal of this water led to both the loss of biological activity [18] and the degeneration of specific terahertz transmission spectra [28]. The conformational mobility of water-containing nanoclusters in these quasi-crystalline preparations (detected by phonon surface vibrations [17] and absorption fluctuations in the near-IR range [29]) was most likely the cause of the thermal radio emission. It should be noted that the maximum values of the microwave radiation flux were observed specifically for drugs with a maximum drug activity (the virus-neutralizing antibodies and the AINF-γ based drug, № P (000023)-(RG-RU)-140422).

The demonstrated specificity of radio emission of nanoparticles in aqueous solutions and as components of amorphous powders (Figure 1 and Table 3 and Table 4) suggests the potential of using the integral thermal radio emission to expressly control the stability of drugs, primarily vaccines. With such prospects, some characteristics of the phenomenon of nanoparticle thermal radio emission required additional study: stability, thermal radio emission, concentration dependence, and relaxation stability. All these parameters are important to further validate the new method for the remote quality control of drugs containing nanoparticles.

### 3.4. Stability of Nanoparticle Thermal Radio Emission in Storage

The time stability of the thermal radio emission of the nanoparticles with drug activity was demonstrated using the AINF-γ preparation under heat activation and the SARS-CoV-2 VLP preparation under light activation as examples. Figure 3 shows that for almost 300 days, the average values of the thermal radio emission flux density almost did not change for the AINF-γ powdered preparation (curve 1). For a vaccine preparation of virus-like particles (which had successfully passed Phase 1 clinical trials by the time this article was written), the thermal radio emission signal was stable for at least 7 weeks (see the graph in the inset). It is very important to emphasize that for almost one year, the placebo and excipient preparations were also controlled (Curves 2 and 3), emphasizing the instrumental stability of the TES92-based unit used and the absence of significant electromagnetic interference for the entire period of the one-year experiment.

### 3.5. Concentration Dependence of Nanoparticle Thermal Radio Emission in Storage

To demonstrate the concentration dependence, a preparation of silver proteinate in paraffin was selected. This model ruled out evaporation and therefore reduced all interference associated with heat dissipation in an aqueous solution to zero. This model also suppresses the Brownian motion of nanoparticles, as well as their transfer in the dish with “surface-bottom” convection flows [30]. This ruled out many possible artifacts that could also generate thermal radio emission in the gigahertz and sub-terahertz ranges [31].

Following from Figure 4, the flux density of the thermal radio emission increased monotonically with an increase in the concentration of silver proteinate in paraffin, reaching a saturation of approximately 0.1% (in terms of Ag). Exposure of the preparation to blue light with a high-power flux density led to a multiple amplification of the signal, which experimentally confirmed Equation (3) and the proposed nanoantenna model. Interestingly, the intrinsic microwave emission of Ag-containing nanoparticles could be the cause of the “remote” antibacterial activity of surfaces in which silver is “frozen” into the solid layer of protective antibacterial coating. This is in good agreement with the model of the effect of gyrotron radiation on infected cell cultures [7].

### 3.6. Relaxation Characteristics of Thermal Radio Emission from a Solution of Nanoparticles

During the energy “pumping” of the nanoantenna, the relaxation times from the excited state should be shown experimentally, as demonstrated by the curve in Figure 5. This indicates that the increase in the average maximum value of the thermal radio emission density took no longer than 5 min, and the relaxation to the initial state after turning off the light took no longer than 10 min for the aqueous solution of humic–fulvic acids. The obtained relaxation characteristics confirm the correctness of the selected measurement technique for a period of 30–40 min (that is, several times longer than the characteristic relaxation times).

The transition of a nanoparticle (NP) to the activated state (NP*) could be described by a minimal, reversible (quasi-equilibrium) kinetic scheme:(6)NP→+hν    k+←−hν    k−NP*

In the presented scheme (6), k_+_ and k_−_ are the constants of the activation/deactivation rate of the nanoparticle. Taking into account the kinetics in Figure 5 and the simplest Eigen relaxation model, the characteristic relaxation times are τ_r_ = (k_+_ + k_−_)/k_+_k_−_ > 5 min. This forced or spontaneous conformational transition when additional energy is captured by the nanoantenna indicates that the supramolecular vibrations of the entire nanoparticle or its domains are the source of radio emission. One study [32] drew attention to the fact that polar dielectrics (which could include all polypeptide nanoparticles in this article) support surface phonon polariton modes in the infrared (IR) frequency range, which are similar to the surface plasmon modes supported by metals in the visible range. As a result, dielectric antennas can be designed to have infrared absorption cross sections with amplification of resonant phonon-polaritons. Consequently, the wavelengths of emissions from biologically active nanoparticles can be much larger than their sizes, up to the millimeter range. To confirm this assumption, we used the filter method in the unit.

### 3.7. In What Spectral Range Do Nanoparticles Emit?

Since nanoparticles, as polar dielectrics, support surface phonon-polariton modes in the IR frequency range, it would be expected that the classical concepts of antenna theory are not applicable to them. In particular, the structural elements of a nanoantenna are not scalable proportionally to the emission wavelength (compared to the classical theory of antennas) due to resonance effects. The nanoantenna size is determined by a parameter similar to the plasma frequency in a metal [33]. Thus, an emitting nanoantenna converts the energy of the localized field into the emission energy in the far field [34]. Similar to optical antennas [33] or nanosized IR-terahertz converters [35], the investigated nanoparticles emit in the microwave band. The problem formulated in this article did not claim more than to distinguish whether thermal radio emission was in the SHF (3–30 GHz) or EHF (30–300 GHz) bands.

Plates or films of various materials were installed in the measuring unit between the detector and the sensor so that their area was ten times larger than the area of the Petri dish containing the emitting solution of INF-α nanoparticles. Figure 6 shows the kinetics of a set of average maximum values of thermal radio emission with various thin screens. Copper, copper–nickel (radio-shielding fabric), and steel screens completely suppressed the signal on the detector, as expected. More than 90% of the signal was shielded on the Parafilm film (consisting mainly of waxes). A shielding of nearly 70% was provided by a 1mm polystyrene layer. This indicates that more than two-thirds of the signal strength are in the EHF band. Moreover, the available data on the attenuation spectra of millimeter waves in the atmosphere and aerosol medium [36,37] indicate that the main emission power of nanoparticle solutions should be expected to be concentrated in the frequency range above 75 GHz (the W band and higher frequencies).

To test this assumption, we investigated the spectra of powdered preparations using a microwave spectrometer in the frequency range of 68 GHz–111 GHz (Table 5). We did not find any specific peaks, but the AINF-γ radio-emitting preparation was significantly different in the median values and signal-to-noise characteristics. Such data may either be a characteristic of the emitter or may reflect insufficient spectral sensitivity of the instrument, which depends on the combination of the spectrometer and horn antenna with a preamplifier.

## 4. Discussion

The main result of this work is the detection of the microwave emission of nanoparticles. This is true for both aqueous solutions and powdered amorphous substances with a high percentage of water (the lactose preparations used in the article are characterized by an 8% moisture content). It is very important that the microwave emission from solutions of aqueous, spherical latex nanoparticles of various diameters or surfactant micelles is not different from the ambient thermal radio background (Table 2). This control is critical in the absence of anechoic chamber conditions and the possible occurrence of sporadic artifacts. From our point of view, many months of observation of the stability of the results obtained, which presented for the drug and placebo at the same time (Figure 3), confirm the feasibility of the experiment without using an anechoic chamber, subject to the strict reproducibility of the unit geometry and the control of the absence of random interference (due to emissions from modems, smartphones, other units, etc.).

The emission of nanoparticles in the microwave range exceeding the background values for a given temperature implies, first, the presence of an energy source (heating or lighting (Figure 2)) and second, the possibility of the slow transformation of the energy into thermal radio emission with rather slow relaxation kinetics (see Figure 5 and Equation (5)). On this basis, we propose the following model for the emitting nanoantenna.

### 4.1. Qualitative Model of the Emitting Nanoparticle

A nanoparticle in an aqueous solution enters into Van der Waals “surface-to-surface” interactions [38] with nonequilibrium density heterogeneities similar to the interaction of a swelling polymer with water clusters (bubstons [39]). Slow relaxation (see [39] and Equation (4)) is caused by the conformational mobility of nanoparticles, which is well described for polypeptides and, in the case of enzymes, is expressed in hysteresis effects during catalysis [40,41]. In addition to the classical fluctuation, the dipole moment on the nanoparticle and the field it creates can be long-lived due to the electrolytic dissociation of ionogenic groups. The contribution to the effective polarizability from the rotation of the nanoparticle (the rotation of its dipole moment) is very significant for a polar liquid. For orientational fluctuations of the dipole moments of molecules in the polar liquid, the characteristic frequencies belong to the microwave region of the spectrum [42]. The polarization of an irregularly shaped, dispersed phase particle with sizes below (l) can lead to the appearance of a potential up to 200 mV (Δϕ), which will result in an electric field strength of E = Δϕ/l over 2 × 10^6^ V/cm. Such values of the electric field strength contribute to the formation of quasi-plasma objects from supramolecular electrons. Cold plasma emission in a wide frequency range (including Rietberg resonances), including the microwave region of the spectrum, is well known [43].

The formation process of an emitting quasi-plasma nanoantenna on a nanoparticle in the polar liquid can be described by the following kinetic scheme (7):(7)NP→+Energy    k+←k−   fast stage   NP*→  slow  ←  fast  NP#

After the assimilation of the “nanophase” NP energy particle, a metastable, activated-state NP* is generated. This can either quickly relax to the initial, non-activated state or undergo a slow transition to the NP^#^ state, accompanied by dipole stabilization and the generation of a quasi-plasma region.

Kinetic scheme (7) is feasible under the following conditions:The nanoparticle is capable of reversible conformational transitions stimulated by an external energy source. This is a well-known property of all polypeptides and oligomeric protein molecules. In this case, a slow transition to a plasma-like object can be similar to non-monotonic transitions in semiconductors of microwave emitters (such as a Gunn diode or higher-frequency modern emitters [44]);The shape of the nanoparticle is a complex, nonconvex polyhedron which contributes to the stabilization of the plasma region and conditions of the extremely heterogeneous distribution of the electric field on “sharped” surfaces. For this reason, we recorded excess emissions, compared to the background, from particles with a very complex surface shape (Table 3).

It should be emphasized that the interaction of nanoparticle surfaces through thermal radio emission can be one of the natural, decisive factors in the synchronization of many processes in organs and tissues, primarily during ligand–receptor interactions, interactions of organelles, and the interactions of cells.

### 4.2. Applications of the New Method for Problems of Pharmaceutical Analytical Chemistry and Biopharmaceutical Analysis

It should be noted that with a sufficient intensity of the described thermal radio emission and the selection of specifically activated nanoparticles, the proposed method of investigating aqueous solutions can be very promising for the express control of immunobiological preparations (vaccines), including without opening the package. From our point of view, the proposed approach is also interesting for developing a method for the non-invasive control of a number of nanoparticles (including specific antibodies) in blood plasma in which the patient has to put their palm [on the device] to determine specific nanocomponents in their tissues. With these prospects in mind, in the conclusion of the article, we would like to summarize some validation characteristics of the method obtained in this article.

## 5. Conclusions

In this paper, a new phenomenon of the specifically induced thermal radio emission of nanoparticles with complex shapes was described. This could serve as the basis for a new quality-control method for biologically active nanoparticles, including vaccines and antibodies. This method of biopharmaceutical analysis is characterized by the following validation parameters:Specificity. Demonstrated on various preparations of mAb to SARS-CoV2 (Table 3) and preparations of various chemical natures (Figure 2, Table 4), the specificity is achieved, among other things, through the use of various methods of activation of thermal radio emission;The limit of detection is up to 0.1 μg in a sample (Table 3—10% of the measured signal);Monotonic dependence on the concentration (Figure 4) that obeys the Van’t Hoff law;Annual measurement stability at an error of 10–12% (Figure 3).

The proposed QA/QC characteristics allow the new method to be considered an original addition to the work that uses metal nanoparticles to solve analytical and pharmaceutical problems [45,46,47].

## Figures and Tables

**Figure 1 pharmaceutics-15-00966-f001:**
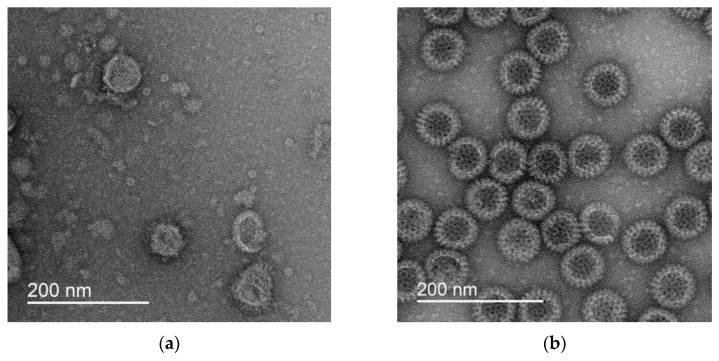
Photomicrographs of VLPs mimicking SARS-CoV-2 (**a**) and rotavirus A (**b**).

**Figure 2 pharmaceutics-15-00966-f002:**
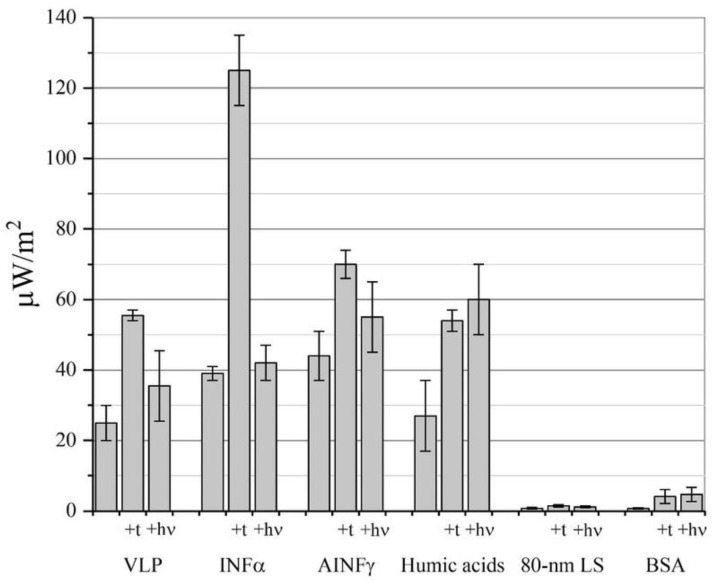
The power density of thermal radio emission of nanoparticle solutions (VLP of rotavirus, INF-α, humic acids, and latex spheres (LS)) and powders containing nanoparticles (AINF-γ and BSA). The data are shown at 23 °C and 37 °C (+t) and irradiation at λ = 412 nm (+hν) (SD, *n* = 9, *p* > 0.95).

**Figure 3 pharmaceutics-15-00966-f003:**
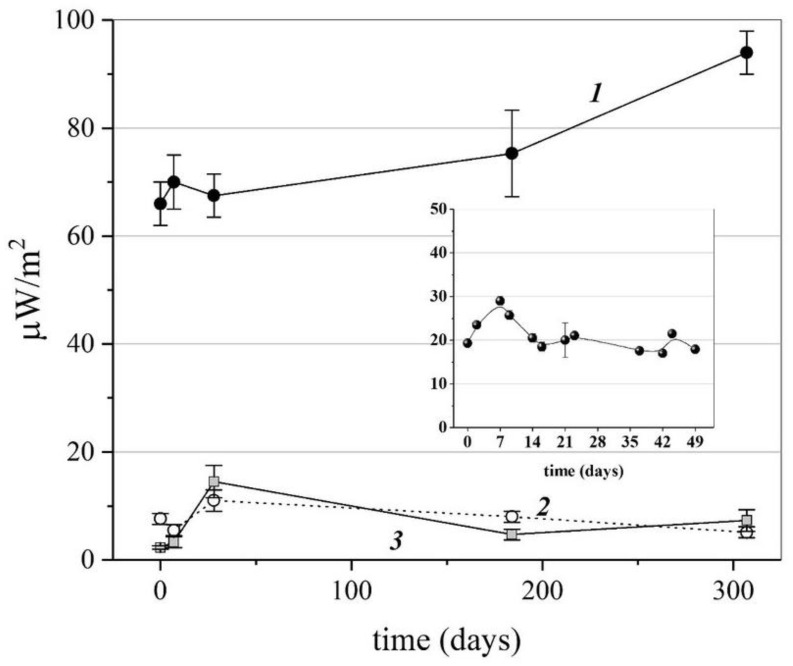
Time stability of thermal radio emission during heat activation of the AINF-γ preparation (t = 37 °C, Curve 1) and light activation of SARS-CoV-2 VLP (λ = 412 nm, see the inset). Curves 2 and 3–thermal radio emission of lactose powder and placebo to the AINF-γ drug preparation, respectively.

**Figure 4 pharmaceutics-15-00966-f004:**
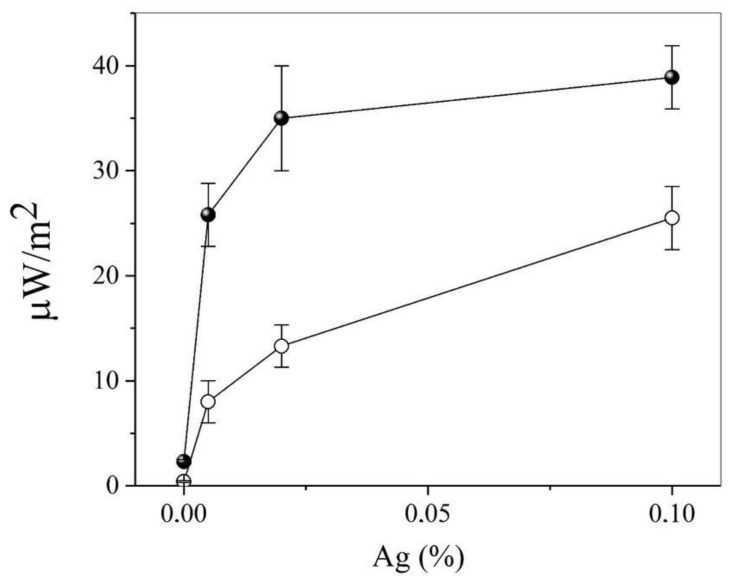
Dependence of the thermal radio emission flux density on the concentration of silver proteinate nanoparticles (the drug concentration is indicated in terms of elemental silver) in a 1 mm layer of paraffin at t = 23 °C (white circles) and t = 37 °C (dark circles–balls).

**Figure 5 pharmaceutics-15-00966-f005:**
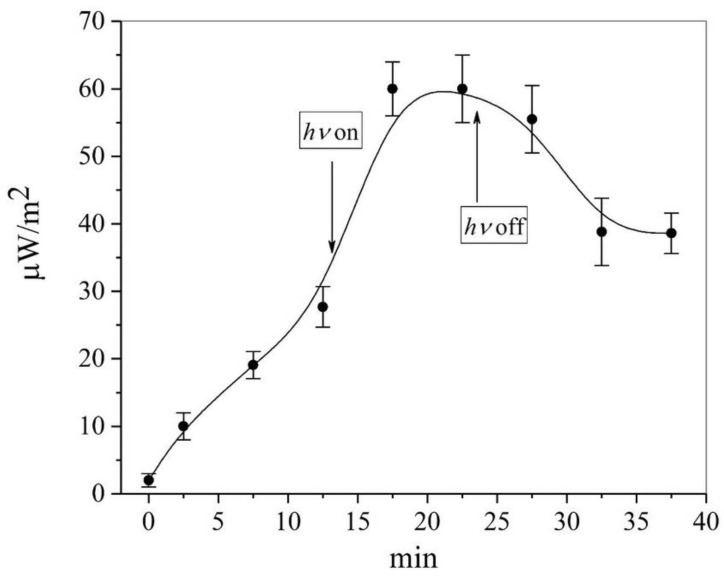
Kinetics of increase and relaxation of the flux density of the thermal radio emission of humic–fulvic nanoparticles. The arrows indicate the light on and off at 412 nm.

**Figure 6 pharmaceutics-15-00966-f006:**
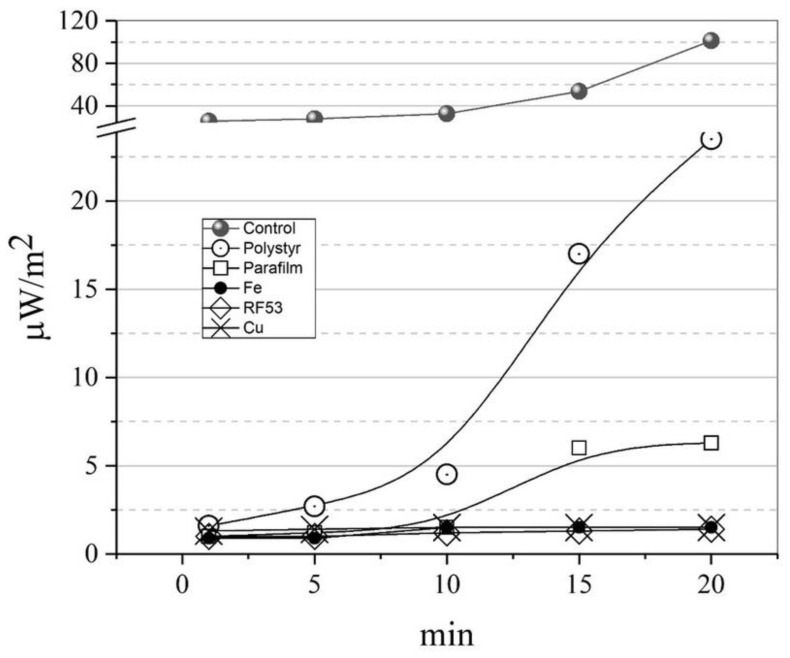
Shielding of the thermal radio emission of the INF-α preparation with screens made of polystyrene (labelled “Polystyr” on the graph; thickness—1 mm), parafilm (“Parafilm”; 1 mm), steel (“Fe”; 5 mm), fabric with copper–nickel particles “RF53” (1 layer), copper grid (“Cu”; 0.2 mm; cell pitch—1 mm).

**Table 1 pharmaceutics-15-00966-t001:** Characterization of monoclonal antibodies.

Clone	Antigen	Interaction with Recombinant Antigen (ELISA)	Interaction with Inactivated Virus (ELISA)	Neutralization of the Virus In Vitro
1G2	RBD	**+**	Not determined	-
3D3	RBD	**+**	Not determined	-
2E7	RBD	**+**	Not determined	-
2E1	RBD	**+**	**+**	**+**

**Table 2 pharmaceutics-15-00966-t002:** **The thermal radio emission of nanoparticle solutions depends on their shape.** The aliquot was 100 µL of aqueous solutions of nanoparticles. The concentration of nanoparticles in aqueous solution was 10 µg/mL. The concentration of silver proteinate in paraffin was 0.05% in terms of the Ag content. The concentration of humic–fulvic acids was 73 mg of dry residue in 1 mL.

EMR Flux Density in the Radio Range According to TES92 Data (t = 37 °C), µW/m^2^					
Spherical Nanoparticles			Complex Shape Nanoparticles		
Latex spheres					
20 nm		**2.6 ± 1.4**	mAb to RBD of SARS-CoV-2		
40 nm		**1.4 ± 0.9**		7 nm	**28 ± 3**
60 nm		**2.6 ± 0.6**			
80 nm		**1.2 ± 0.3**	VLP of SARS-CoV-2	164 nm	**21 ± 4**
100 nm		**1.6 ± 0.4**	VLP of rotavirus	95 nm	**55 ± 2**
BSA	7 nm	**1.0 ± 0.1**	IFNα	4 nm	**125 ± 10**
Micellar water, Faberlic	9 nm	**0.4 ± 0.1**	Humic acids	200 nm	**54 ± 3**
Micellar water, BioDerma	10 nm	**0.5 ± 0.2**	Silver proteinate	60 nm	**38 ± 2**

**Table 3 pharmaceutics-15-00966-t003:** Specificity of thermal radio emission activation using the example of monoclonal antibodies to various RBD epitopes of the SARS-CoV-2 virus. The aliquot of each drug was 100 µL. The concentration of antibodies was 10 µg/mL. 100% EMR flux density corresponds to 28 ± 3 μW/m^2^ (t = 23 °C) for 2E1 * antibodies (without lighting, t = 23 °C) with the background (control, t = 23 °C) of 1.3 ± 0.4 μW/ m^2^.

Preparation	EMR Flux Density in the Radio Range According to TES92 Data, %	
	Heating to 37 °C	Lighting, λ = 412 nm
Control (Phosphate-Buffered Saline)	5	3
2E1 *	99	100
1G2	42	92
2E7	73	27
3D3	55	9

*—virus-neutralizing antibodies (Table 1).

**Table 4 pharmaceutics-15-00966-t004:** The specificity of activation of thermal radio emission using the example of powdered preparation № P (000023)-(RG-RU)-140422 (AINF-γ) and its placebo, artificially prepared in the fluidized bed chamber (Placebo–the active pharmaceutical ingredient was replaced by a water–alcohol solution in the aerosol chamber). A total of 30 g of the drug was poured into the Petri dish of the TES92 unit solution. A 100% EMR flux density corresponds to 44 ± 7 µW/m^2^ for INF-γ.

Preparation	EMR Flux Density in the Radio Range According to TES92 Data, %	
	23 °C	37 °C
AINFγ	100	159
Placebo	25	11
Lactose (powder)	16	18

**Table 5 pharmaceutics-15-00966-t005:** Average characteristics of the intensity of the intrinsic radio emission of preparations in the frequency range of 68 GHz–111 GHz.

	AINF-γ	Placebo	Lactose (Powder)	Background (Empty Dish)
S/N *	330	290	290	200
(M + 100) × 1000, dBm **	42	−271	−352	−505

* S—intensity (amplitude) of emission (dBm) averaged over the entire frequency range, N—standard deviation over the entire sample of amplitudes (*n* = 1001); ** M—median values of amplitudes.

## Data Availability

Not applicable.

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
