# Peer review of "Radiothermal Emission of Nanoparticles with a Complex Shape as a Tool for the Quality Control of Pharmaceuticals Containing Biologically Active Nanoparticles"

_pharmaceutics, 2023, doi:10.3390/pharmaceutics15030966_

Round 1

Reviewer 1 Report

1. The title of the article - Development of new method for quality control of pharmaceuticals containing biologically active nanoparticles - is too exaggerated and does not correspond to the study carried out and the results obtained

2. This issue of the journal is dedicated to the features of supramolecular drugs - what is the role of this sentence in the presented study?

3. Raw 73: was obtained as follows. Cell strains – correct the sentences

4. Raw 84: by the standard method in the cooled growth medium  without serum – mention the reference

5. Raw 123: Obtaining and purification of virus-like particles (VLP) mimicking SARS-CoV-2. – this represent a sub-title?

6. Raw 153: Sistema Biotechnologii – is correct?

7. Raw 155: Lyophilized preparation of bovine serum albumin was received from the Sigma-Aldrich company – correct the sentence

6. Sub-chapter 2.1. Nanoparticles – each nanoparticle used in study must be explained from the viewpoint of the reason of use, as well as clearly presented the preparation  

7. All abbreviations must be explained

8. reformulate the phrase as it is not clear enough: For example, for particles with sizes r < 100 nm, it was established [20] (with a combination of Kirchhoff's law and Mie scattering theory, taking into account the absorption cross section of equilibrium electromagnetic radiation  by dipole particles) that the intensity of their thermal radiation I, normalized to black body  radiation Ib

9. It is not clear in Figure 2 data corresponding to the determinations made at 23°Ð¡ or 37°Ð¡

10. Table 2 Is not clear presented

11. Raw 339: of the AINF powdered preparationg, comparing the results based of the thermal – correct

12. as stated the authors: with a sufficient intensity – not a specific value - of the described thermal radio emission and selection of specifically activated nanoparticles – and not generally valid for any nanoparticle- , the proposed method of investigating aqueous solutions can be very promising for express control of immunobiological  preparations (vaccines), including without opening the package.

13. the study demonstrated specificity of radio emission of nanoparticles in aqueous solutions and as components of amorphous powders and from this viewpoint the manuscript is interesting

Author Response

  1. The title of the article - Development of new method for quality control of pharmaceuticals containing biologically active nanoparticles - is too exaggerated and does not correspond to the study carried out and the results obtained. We changed the title: 

    Radiothermal emission of nanoparticles with complex shape as a tool for quality control of pharmaceuticals containing biologically active nanoparticles

  2. This issue of the journal is dedicated to the features of supramolecular drugs - what is the role of this sentence in the presented study? 

    Raw 50: The first line in the introduction was omitted.

  3. Raw73: was obtained as follows. Cell strains – correct the sentences. 

    Raw 81:… was obtained in the manner described below. Cell cultures (specific cell strains) …

  4. Raw 84: by the standard method in the cooled growth mediumwithout serum – mention the reference 

    Raw 93: [15]

  5. Raw 123: Obtaining and purification of virus-like particles (VLP) mimicking SARS-CoV-2. – this represent a sub-title? Raw 138: It was changed.
  6. Raw 153: Sistema Biotechnologii – is correct? 

    Yes, it is transliteration from Russian.

  7. Raw 155: Lyophilized preparation of bovine serum albumin was received from the Sigma-Aldrich company – correct the sentence. 

    Raw 172: Bovine serum albumin (lyophilized powder) was obtained from Sigma-Aldrich company (USA).

  8. Sub-chapter 2.1. Nanoparticles – each nanoparticle used in study must be explained from the viewpoint of the reason of use, as well as clearly presented the preparation. 

    A comment:

    The nanoparticles studied in this article were selected from a complex of pharmaceuticals and auxiliary preparations developed as part of anti-epidemic actions and tests at the Gamaleya Center of the Ministry of Health of the Russian Federation and the Ministry of Health of Moscow during the SARS-CoV-2 coronavirus pandemic. Among them are antibodies to SARS-CoV-2, virus-like particles (VLP) of SARS-CoV-2 and rotavirus, pharmaceuticals based on antibodies to interferon gamma, a complex of humic-fulvic acids. Antibodies to SARS-CoV-2 were obtained against different epitomes, which made it possible to demonstrate the specificity of the new method. Antibodies, and VLP, and complexes of humic-fulvic acids, and interferons are nanoparticles of a very complex shape. As a kind of control, we took biologically inert nanoparticles with a shape close to spherical: latex spheres and bovine serum albumin.

  9. All abbreviations must be explained 

    Missing abbreviations are added:

    Raw 80: RBD (receptor-binding domain)

    Raw 98:  Hypoxanthine-Aminopterin-Thymidin (HAT) media supplement

  10. reformulate the phrase as it is not clear enough:For example, for particles with sizes r < 100 nm, it was established [20] (with a combination of Kirchhoff's law and Mie scattering theory, taking into account the absorption cross section of equilibrium electromagnetic radiation by dipole particles) that the intensity of their thermal radiation I, normalized to black body  radiation Ib 

    Raw 271: Corrected version for the sentence: For example, for particles with sizes r < 100 nm, it was established [20] (based on a combination of Kirchhoff's law and Mie theory), that the intensity of their thermal radiation I, normalized to black body radiation Ib

  11. It is not clear inFigure 2 data corresponding to the determinations made at 23°Ð¡ or 37°Ð¡ 

    A comment:

    The first column (23°Ð¡) corresponds to column without sign. The second column (37°Ð¡) corresponds to column with sign (+t). The third column (irradiation at l=412 nm) corresponds to column with sign (+hn).
  12. Table 2 Is not clear presented. 

    It was changed.

  13. Raw 339: of the AINF powdered preparationg, comparing the results based of the thermal – correct. 

    Raw 393: Corrected version for the sentence:

    We assumed that this approach would be universal and applied it to the AINF powders. Radiothermal radiation (with thermal and light activation) for the drug and its placebo was compared. Moreover, the placebo was specially made according to our order in a fluidized bed aerosol chamber, where the AINF preparation is also produced (Table 4).

  14. as stated the authors: with a sufficient intensity– not a specific value - of the described thermal radio emission and selection of specifically activated nanoparticles – and not generally valid for any nanoparticle- , the proposed method of investigating aqueous solutions can be very promising for express control of immunobiological  preparations (vaccines), including without opening the package. 

    A comment:

    We have emphasized the term "sufficient intensity", meaning that during operation without opening the package, part of the thermal radio emission will be scattered by the material of the package walls. The specificity of radiation for each drug must be proved taking into account the tasks to be solved. For example, a manufacturer is confident in the authenticity of a drug and needs express control of nanoparticle nativeness and coagulation stability. Solving the problems of determining authenticity in the most general form requires, of course, the study of the radio emission spectrum.

  15. the study demonstrated specificity of radio emission of nanoparticles in aqueous solutions and as components of amorphous powders and from this viewpoint the manuscript is interesting 

    A comment:

    We thank the reviewer for all his comments and tried to make all corrections.

Reviewer 2 Report

The authors report the development of a new method for the quality control of pharmaceutical products containing biologically active nanoparticles, the quality of the data is good and the discussion is cogent. Some minor corrections are recommended 

Title: OK

Abstract:

1) line 23 "we studied the intrinsic" writing in third person. 

2) line 34 "We assumed" writing in third person

Keywords: OK

Introduction: The authors present a good contextualization of the subject, the knowledge gap is clear and the formalization of the research allows the reader to be interested in the manuscript. 

1) Remove this line "This issue of the journal is dedicated to the features of supramolecular drugs"

2) Line 59 "We assumed" writing in third person.

3) Line 66 "we focused our" writing in third person

Materials and Methods: The methodology is clear, well detailed and would allow the procedure to be replicated in other laboratories. 

Results: the results are presented in a clear manner, the quality of the data makes it possible to understand the results.  

1) Line 256 "We found" writing in third person. 

2) line 259: the expressions "logical" and "surprising" can be understood as hyperboles. It is recommended to replace the words.

3) Line 266 "our point". writing in third person

4) Line 298 "we have" writing in the third person 

5) Line 316 "Obviously," is a hyperbole. Nothing in research is obvious

6) Line 324 "we demonstrated" writing in the third person 

7) Line 338 "We assumed" "we applied" writing in the third person 

8) Line 366 "We demonstrated" writing in the third person 

9) Line 383 "we chose" writing in the third person 

Discussion: Returning to the quality and clarity of the reported results, the discussion of the results is clear and forceful. The discussion highlights the importance of the research and the practical potential of the results. 

Conclusions:  writing in the third person 

Author Response

We would like to thank the reviewer for all valuable comments and remarks which helps us to really improve the scientific quality of our revised manuscript. Details of our responses to each reviewer comment are shown below. We hope you will find these revisions rise to your expectations.

Abstract:

1) line 23 "we studied the intrinsic" writing in third person. Line 24 changed "has been studied". 

2) line 34 "We assumed" writing in third person. Line 35 changed "It was assumed".

Introduction

1) Remove this line "This issue of the journal is dedicated to the features of supramolecular drugs". 

The first line in the introduction was omitted.

2) Line 59 "We assumed" writing in third person. Line 66 changed "It was assumed".

3) Line 66 "we focused our" writing in third person. Line 73 changed "it is important to pay attention".

Results:

1) Line 256 "We found" writing in third person. Line 279 changed "It was found that". 

2) line 259: the expressions "logical" and "surprising" can be understood as hyperboles. It is recommended to replace the words. Line 282 changed "would be expected".

3) Line 266 "our point". writing in third person. Line 289 deleted.

4) Line 298 "we have" writing in the third person. Line 331 changed and moved "were separated". 

5) Line 316 "Obviously," is a hyperbole. Nothing in research is obvious. Line 355 deleted "Obviously,".

6) Line 324 "we demonstrated" writing in the third person. Line 365 deleted "we" and moved. 

7) Line 338 "We assumed" "we applied" writing in the third person. Line 393 changed  "It was assumed", "and applied". 

8) Line 366 "We demonstrated" writing in the third person. Line 423 changed and moved  "was demonstrated". 

9) Line 383 "we chose" writing in the third person. Line 455 changed and moved "was selected".

Conclusions:  writing in the third person. 

Line 627 changed.

Reviewer 3 Report

In this paper, we described a new phenomenon of specifically induced thermal radio 538

emission of nanoparticles with complex shapes, which could serve as the basis for a new quality control method of biologically active nanoparticles, including vaccines and anti-bodies.

The paper is well designed and well written and carries a novel idea, so I recommend the acceptance of the manuscript after performing the following:

1- The first line in the introduction should be omitted.

2- Line 123: Is this a title "Obtaining and purification of virus-like particles (VLP) mimicking SARS-CoV-2" ? If not please revise the statement.

3- The nature of error bars in Figure 2 needs to be stated in the figure caption.

Author Response

We would like to thank the reviewer for all valuable comments and remarks which helps us to really improve the scientific quality of our revised manuscript. Details of our responses to each reviewer comment are shown below. We hope you will find these revisions rise to your expectations.

1 - The first line in the introduction was omitted.

2 - Line 123: Is this a title "Obtaining and purification of virus-like particles (VLP) mimicking SARS-CoV-2" ? If not please revise the statement. Line 138: Revised the statement.

3 – The nature of error bars in Figure 2 needs to be stated in the figure caption. The nature of error bars is Standard Deviation (n=9, P>0.95).

Round 2

Reviewer 1 Report

the revised version of the manuscript can be published